# The Real Impact of Age on Mortality in Critically Ill COVID-19 Patients

**DOI:** 10.3390/jpm13060908

**Published:** 2023-05-29

**Authors:** Agamemnon Bakakos, Evangelia Koukaki, Sevasti Ampelioti, Iliana Ioannidou, Andriana I. Papaioannou, Konstantinos Loverdos, Antonia Koutsoukou, Nikoleta Rovina

**Affiliations:** 1st University Department of Respiratory Medicine, National and Kapodistrian University of Athens, 11527 Athens, Greece; e.koukaki@yahoo.gr (E.K.); sevi.ampelioti@gmail.com (S.A.); ilianaioan@hotmail.com (I.I.); papaioannouandriana@gmail.com (A.I.P.); kloverdos@yahoo.com (K.L.); koutsoukou@yahoo.gr (A.K.); nikrovina@uoa.gr (N.R.)

**Keywords:** SARS-CoV-2, ICU, elderly, COVID-19, respiratory failure, age

## Abstract

Objective: The impact of severe infection from COVID-19 and the resulting need for life support in an ICU environment is a fact that caused immense pressure in healthcare systems around the globe. Accordingly, elderly people faced multiple challenges, especially after admission to the ICU. On this basis, we performed this study to assess the impact of age on COVID-19 mortality in critically ill patients. Materials and Methods: In this retrospective study, we collected data from 300 patients who were hospitalized in the ICU of a Greek respiratory hospital. We split patients into two age groups using a threshold of 65 years old. The primary objective of the study was the survival of patients in a follow up period of 60 days after their admission to the ICU. Secondary objectives were to determine whether mortality is affected by other factors, including sepsis and clinical and laboratory factors, Charlson Comorbidity Index (CCI), APACHE II and d-dimers, CRP, etc. Results: The survival of all patients in the ICU was 75.7%. Those in the <65 years old age group expressed a survival rate of 89.3%, whereas those in the ≥65 years old age group had a survival rate of 58% (*p*-value < 0.001). In the multivariate Cox regression, the presence of sepsis and an increased CCI were independent predictors of mortality in 60 days (*p*-value < 0.001), while the age group did not maintain its statistical significance (*p*-value = 0.320). Conclusions: Age alone as a simple number is not capable of predicting mortality in patients with severe COVID-19 in the ICU. We must use more composite clinical markers that may better reflect the biological age of patients, such as CCI. Moreover, the effective control of infections in the ICU is of utmost importance for the survival of patients, since avoiding septic complications can drastically improve the prognosis of all patients, regardless of age.

## 1. Introduction

In December 2019, the first cases of pneumonia that led to severe respiratory failure were described in Wuhan, China; these were later attributed to the newly discovered SARS-CoV-2 virus. The spread of the virus was extremely rapid, leading the World Health Organization (WHO) to declare a pandemic only three months later, in March 2020. Currently, infection from SARS-CoV-2 mostly leads to mild symptoms such as cough, fever, and fatigue; however, it can still cause severe infection, eventually leading to severe bilateral pneumonia and Acute Respiratory Distress Syndrome (ARDS), requiring support in the ICU [1].

It is now established that the likelihood of severe infection increases with age, the presence of comorbidities, and, of course, the absence of immunization against SARS-CoV-2. Although most critically ill patients suffer from severe type I respiratory failure due to the development of ARDS, there are several important complications that can occur during their stay in the ICU. The most predominant are barotrauma due to mechanical ventilation, coagulation disorders, such as pulmonary embolism, abdominal involvement, such as acute kidney failure, and mesenteric ischemia. Thus, patients require a holistic approach, since extrapulmonary complications can greatly affect the prognosis of patients in the ICU environment [2,3].

Several comorbidities were associated with severe infection and a worse prognosis (increased mortality, days of hospitalization, need for mechanical ventilation, septic complications; this was clearly demonstrated in the very old intensive care patients (COVIP) group study that evaluated the true burden of COVID-19 infection in the elderly population (age > 70 years old). With a total number of 1346 patients (72% males) and a median age of 75 years old, researchers split patients into three groups based on clinical frailty score (CFS)–fit, vulnerable, or frail–and observed 30-days mortality. Total survival was estimated at 59%. Those who were deemed fit exhibited a survival rate of 66%, the vulnerable group showed a survival rate of 53%, and finally the frail group had a survival rate of 41%, with all *p*-values < 0.001. Researchers further split the patients belonging to each group into three sub-groups: the 70–80, the 80–90, and the 90+ year-old group. There was no statistically significant difference observed in mortality rates in patients in the same CFS group, despite their age difference. In the multivariate analysis, belonging in the frail group was deemed to be an independent factor affecting mortality [4]. These findings raise the question as to whether age alone is enough to guide clinical decision making, such as which patients should be supported with mechanical ventilation in an ICU environment in times of crisis where beds in the ICU are not abundant. 

The next question that arises is whether there is a biomarker or a combination of biomarkers that can predict the likelihood of severe infection and/or the survival of patients in the ICU. In another study of the COVIP group, lactic acid values were used to predict the mortality of patients; values < 2 mmol/L were linked to better prognosis in all age groups [5]. The predictive value of other biomarkers, including CRP, d-dimers, neutrophil to lymphocyte ratio (NLR), has also been evaluated in numerous studies; there are no clear results that a single biomarker can accurately predict severe infection or increased mortality [6].

Studying literature concerned with SARS-CoV-2, it is clear that elderly patients are not adequately represented and therefore the true burden of the infection in these patients has not yet been elucidated. Elderly patients usually suffer from more comorbidities when compared to younger patients and the therapeutic challenge for healthcare providers is great, since a mild viral infection can exacerbate chronic health problems, leading to death. In multiple cohorts around the world, it was demonstrated that several comorbidities, including heart failure, diabetes mellitus, chronic obstructive pulmonary disease (COPD), can affect the prognosis of a patient, especially if they co-exist. On the other hand, comorbidities such as bronchial asthma have not yet been linked with worse prognosis when compared to healthy individuals [7,8].

The aim of this original research is to evaluate the true burden of SARS-CoV-2 infection in the elderly population in the ICU, because elderly patients were clearly neglected during the first pandemic waves. In countries that were hit fast and harshly by the spread of COVID-19, decision making as to which patients would be supported in the ICU was sometimes taken solely with age as the criterion. This raises both scientific and ethical questions.

In this retrospective single-center study, which was conducted based on patient records kept online at the ICU department of a Greek respiratory hospital, patients were divided into two age groups, with the threshold used being 65 years old. The primary endpoint was 60 days mortality after admission in the ICU between the two age groups. Secondary endpoints included the efficiency of several biomarkers and clinical scores such as APACHE II and Charlson Comorbidity Index (CCI) in predicting which patients have a higher risk of mortality in the ICU.

## 2. Materials and Methods

This is a retrospective observational study that was conceptualized and realized between June 2022 and February 2023 in a single center of “Sotiria Chest Diseases Hospital”, in the ICU department of the 1st University Department of Respiratory Medicine. Patients included in the study were required to have a positive reverse transcriptase polymerase chain reaction (RT-PCR) test for SARS-CoV-2 before or at their admission to the ICU department. Patients hospitalized from September 2020 to January 2022 were included in this study. Patients who were transferred from another ICU after a prolonged stay there (>48 h) and those whose medical records were unavailable due to technical reasons were excluded from the study. 

All data was retrieved from the medical files kept in the electronic system of the hospital (medico//s of Siemens Medical Solutions). From a total of 319 patients hospitalized during the aforementioned timeframe who were screened for inclusion, 300 were finally included in the study (Figure 1).

Patients were split into two age groups with a threshold of 65 years of age. The threshold of 65 years was used since it is the most widely used criterion to divide elderly patients around the world. It has been used by the WHO and it is also used in the USA and the UK [9,10]. Nevertheless, it should be noted that researchers have also used other thresholds, such as 60 or 70 years of age [11].

The CCI and the APACHE II score were measured within the first twenty-four hours from the patient’s admission to the ICU. In addition, for CCI, we used data found in the medical records of each patient stored electronically, as well as electronic files of medical prescriptions for each patient, which include the specific ICD-10 diagnoses describing each condition. CCI evaluates a vast number of comorbidities, and higher scores are predictors of increased mortality [12].

The APACHE II score was calculated based on the vital signs of each patient on admission, as well as their laboratory tests from the same day, both recorded in their medical files [13]. All laboratory tests were performed in the Sotiria Chest Diseases Hospital Laboratory to ensure that no discrepancies would occur as a result of values from different laboratories.

The presence of sepsis and/or septic shock was determined by signs of tissue hypoperfusion (lactic acid > 2 mmol/L, systolic blood pressure < 90 mmHg) without evidence of hypovolemia combined with an acute increase of at least 2 points in the Sequential Organ Failure Assessment (SOFA) score consequent to signs of infection [14].

### Statistical Analysis

The statistical analysis of all data was performed with the use of the SPSS 23 statistical package (SPSS Inc., Chicago, IL, USA). The normality of distributions was checked using the Kolmogorov–Smirnov test. Data are presented as n (%) for categorical variables, as mean ± SD for normally distributed, and as median (interquartile ranges) for skewed numerical variables. Comparisons between groups were performed using chi-square tests for categorical data, as well as unpaired *t*-tests or Mann–Whitney U-tests for normally distributed or skewed numerical data. Correlations were performed with Spearman’s correlation coefficient. Overall survival time was calculated from admission to the ICU until death. Patients discharged alive from the hospital were censored at the date of exit. Kaplan–Meier estimates were used to describe and visualize the effect of categorical variables.

For the analysis of the primary objective, survival analysis and Cox regression analysis were implemented. In detail, the times to death according to the presence of a characteristic or adverse event was evaluated with Kaplan-Meier survival curves and log-rank tests. Cox regression univariate and multivariate analyses were performed in order to evaluate the influence of each characteristic or score in ICU mortality. Significant confounders evaluated in Cox regression analyses included age, sex, and APACHE II score. Results are presented as hazard ratios (HR) with 95% confidence intervals (CI).

## 3. Results

### 3.1. Demographic Data

The demographic data of all patients included in the study are represented in Table 1. From the 300 patients included in the study, only 29% were females. More specifically, in the age group < 65 years old, the gender distribution was 74% men and 26% women, while in the group > 65 years old the distribution was 67–33%.

The median age of patients in the age group < 65 years was 51.1 years-old, with a standard deviation (SD) of 9.2 years; in the elderly age group, the median age was 72.3 years, with a SD of 6.1 years. The CCI was significantly higher in the elderly age group (3.8 vs. 1.3); this was was expected since we mentioned before that older people tend to suffer from more comorbidities when compared to younger individuals. The same is true for the APACHE II score, which was higher in the elderly group.

#### 3.1.1. Vaccination Status

It is important to mention the vaccination status of patients participating in this study; as vaccines were not available from the start of the enrollment period (September 2020), only a minority of patients were fully vaccinated (two doses of an mRNA vaccine at that time) either by choice or due to the vaccines not being available yet. From the 300 patients, only 12 (4%) were fully vaccinated. 

#### 3.1.2. Smoking Status

In our cohort, one out of two men were non-smokers while the rest were either former or current smokers with >10 pack-years; three out of four women (73.6%) were non-smokers, with the rest being either former or current smokers. There were no significant differences reported between the two age groups; in the total population only 42 patients (14%) were current smokers.

#### 3.1.3. Comorbidities in the ICU

The most common comorbidity in the total population was arterial hypertension (45.3%), with hyperlipidaemia and diabetes mellitus coming second, both with a percentage of 22.3%. Between the two genders the greatest differences are noted in the prevalence of hypothyroidism, with 28.7% of women suffering compared to just 9.9% of men (*p*-value < 0.001). On the contrary, men suffered twice as much from coronary artery disease (CAD), with a percentage of 12.7% compared to 5.7% of women (*p*-value < 0.001). The last notable difference is in terms of BMI, with men expressing a median BMI of 29.8 ± 5.2, while women were more obese, with a median BMI of 32.3 ± 7.8. There were no significant differences in the presence of other comorbidities between the two genders. As to the two age groups, most comorbidities tend to be more frequent in the elderly group. Arterial hypertension was present in 61.1% of the elderly patients, compared to just 33.1% of the younger age group (*p*-value < 0.001); hyperlipidaemia was present in 31.3% versus 15.4% (*p*-value < 0.001); diabetes mellitus was present in 29% versus 17.2% (*p*-value = 0.015); CAD was present in 16% versus 6.5% (*p*-value = 0.008). Hypothyroidism and COPD did not exhibit statistically significant differences between the two age groups. Finally, the median BMI was lower in the elderly group by a non-statistically significant margin (29.8 kg/m^2^ compared to 31.1 kg/m^2^).

### 3.2. 60 Days Mortality in the ICU

The primary endpoint of this research was 60 days all-cause mortality in the ICU between the two age groups. The 60-day survival of patients in the age group < 65 years old was 89.3% (151 out of 169 patients); the older age group expressed a survival rate of 58% (76 out of 131 patients). The cumulative survival of all patients hospitalized in the ICU was 75.7% (227 patients survived). The univariate Cox Regression shows that the younger age group exhibited significantly reduced mortality (*p*-value < 0.001) (Figure 2). The same is true for ages as a constant variate, since once again the younger age is associated with reduced mortality in the univariate analysis.

Nevertheless, in the multivariate Cox regression analysis, neither age as a constant variate nor the age group managed to express statistical significance concerning 60-days mortality in the ICU (*p*-value = 0.320).

In the multivariate Cox Analysis, all the factors that held a *p*-value < 0.20 were included. The CCI index was correlated with a statistically significant *p*-value < 0.001, with increased 60 days-mortality. The Hazard Ratio (HR) was 1.646 with a Confidence Interval (CI) ranging from 1.247–2.175.

Moreover, another crucial factor that was also associated with reduced mortality in the ICU: the absence of septic complications, with a *p*-value < 0.001. The HR was 0.039, with a CI of 0.013–0.119. Investigating bacterial infections, it was demonstrated that from a total of 300 patients, 65 patients (21.7%) had positive blood cultures during their hospitalization, 134 patients (44.8%) had at least one specimen of positive bronchial culture, and 64 patients (21.7%) had at least one specimen of positive urine culture. Concerning fungal infections, 22 (7.3%) had positive cultures for fungi, with the most predominant species isolated being *Aspergillus* spp. and *Candida* spp.

Concerning laboratory values such as CRP, d-dimers, ferritin and others shown in Table 2, none managed to preserve statistical significance in the multivariate analysis.

The antiviral agent “remdesivir” was largely used during the first waves of the pandemic. In the multivariate Cox Regression analysis, remdesivir did not manage to exhibit statistically significant results by a very slight margin (*p*-value = 0.055); however, it is clear that patients who were unable to receive remdesivir (mainly due to severe renal impairment) expressed a trend to higher mortality when compared to those who received the medication, with a HR = 2.458 and a CI ranging from 0.980–6.162.

## 4. Discussion

The primary endpoint of our research was to evaluate the true impact of age in critically ill patients in the ICU. While other factors emerged during the statistical analysis, such as CCI score and sepsis, age was not found to be an independent prognostic factor. Additionally, the use of remdesivir expressed a positive signal that almost reached statistical significance. Finally, none of the laboratory values was found to be an independent prognostic factor.

Age does play a role in the prognosis of COVID-19 infection, but not as a number or as an age group threshold. Instead, since older age usually means a higher number of comorbidities, elderly patients face increased challenges in the ICU because they suffer from more concomitant diseases and thus are more fragile. This comes into accord with the statistical significance of the CCI score in our analysis. The parameters included in the CCI can more accurately reflect the burden of multiple comorbidities and the function of pivotal organs such as the heart, the kidneys, and the liver, thus more accurately depicting the severity of a patient in the ICU [12]. In our multivariate Cox regression, age was shown to be a confounding factor rather than a significant variable on its own. This result can have many interpretations, which will be analyzed in the next few paragraphs.

First, most, if not all, patients who were admitted in the ICU department suffered from moderate to severe ARDS with a median PaO_2_/FiO_2_ (P/F) ratio of 117.36 ± 55.71. There was no statistically significant difference in the P/F ratio between the two age groups, which means that the severity of the disease was similar in both age groups. Age has been associated with higher rates of mortality, greater need for ICU admission, and length of hospital stay; this is because aged people progress more frequently to severe disease [15]. However, when a patient is already in the ICU with severe COVID-19 infection, other factors, such as the presence of multiple comorbidities and complications such as sepsis, are more important than age when predicting a patient’s outcome.

Another possible explanation is that the pressure on healthcare systems in the first waves of the pandemic put doctors and healthcare providers in a tough situation, where they had to face difficult choices at both a scientific and a moral level. Since there were not enough ICU beds in many countries, Greece being one of them, with more than 95% of its ICU beds being covered for months, there is a chance that the older patients who were admitted to the ICU were in a better clinical state than other elderly people who never got a chance to be admitted to the ICU and who perished in the wards. Age has been correlated with increased need for hospitalization, ICU admission, and mortality in many studies, but the true impact of age on mortality is far more complex to elucidate [15,16,17,18].

As already mentioned, the COVIP study, which included patients over 70 years old from 28 different countries, tried to estimate the true burden of age and showed that the clinical status of a patient affects mortality by a statistically important margin, whereas age does not. This research is in accord with our results, showing that age is not the most important factor and that the “biological” state of a patient is far more important in predicting survival [4].

It is widely accepted that septic complications are one of the most common events in severely ill patients and one of the first causes of death worldwide, especially in an ICU environment. In our study, avoiding septic events was shown to drastically improve survival in patients, irrespective of their age or comorbidities. 

In a systematic review and meta-analysis that included 3834 patients from 30 individual studies, the presence of a microbial or fungal infection in patients already suffering from a coronavirus (SARS-CoV-1, MERS, or SARS-CoV-2) was about 7% for hospitalized patients. However, this percentage is doubled when only patients admitted to the ICU are considered, reaching 14%. Septic complications are not rare in the ICU and all preventive measures should be followed in order to reduce their incidence [19].

In another longitudinal study that evaluated patients suffering from severe COVID-19 requiring ICU admission, it was demonstrated that very few patients developed a bacterial infection during the first 48 h after their admission to the ICU (just 5%), but that this percentage quadrupled and reached 20% by the end of their hospitalization in the ICU. Factors that increased the likelihood of a bacterial infection were prolonged stay in the ICU and prolonged need for mechanical ventilation [20]. 

A recently published multicenter retrospective cohort study that included almost 14,000 COVID-19 patients hospitalized between 2020 and 2022 demonstrated that the impact of a bacterial co-infection in the prognosis of patients (both in the ward and the ICU) is higher than previously described risk factors, such as age or individual comorbidities [21]. Additionally, sepsis was deemed as an important prognostic factor in another single center study from Greece, which included patients from all waves of the pandemic. This result is especially important since the population evaluated has many similarities with the population of our study, as both studies were carried out in Greece across the same time period [22].

It is interesting to note a recently published study, where patients suffering from COVID-19 were recruited from the same ICU as our cohort. The study evaluated the burden of fungal infections in those admitted in the ICU. Among those enrolled in the study, 10.7% of patients developed a fungal infection, with the most common pathogens being *Candida albicans* and *Aspergillus* spp. However, unlike bacterial complications, fungal infections did not negatively affect mortality [23]. In another retrospective study on patients suffering from severe COVID-19 in the ICU, the percentage of bacterial or fungal infections reached 51% [24]. It is clear that the great discrepancy between the results worldwide depends on the different protocols used to prevent septic complications, the different facilities available, and the differences in diagnosing and reporting those complications.

The use of remdesivir produced a positive signal in our research. Although not statistically significant, this result shows that remdesivir can indeed act protectively, even in severe SARS-CoV-2 infection. However, it should be noted that since patients who did not receive remdesivir had a contraindication, such as severe renal impairment (eGFR < 30 mL/min) or severe liver impairment (AST levels > 5-times upper limits of normal), this signal could also be attributed to the fact that patients with more severe end organ damage comprised the group that did not receive remdesivir. Thus, these results should be treated with caution.

Finally, another result worthy of further discussion is the disproportionate rate of males in the ICU compared to females; this is in accordance with reports from other researchers around the world. For instance, in a Scandinavian cohort that recruited and evaluated 5471 patients (of whom 49% were males), it was demonstrated that males require admission to the ICU due to severe COVID-19 infection far more often than females’ (27% of males versus 17% of females). Additionally, in the multivariate analysis of that research, male gender was associated with higher mortality per infected person with an odds ratio (OR) of 2.37 with 95% CI 1.22–4.59, showing that males have higher chances of progression to severe disease that eventually requires ICU admission [25].

Results almost identical to our research were reported from another Mediterranean country, Italy. In an observational study including 2378 patients infected with SARS-CoV-2, hospitalized in 26 different hospitals, of whom 395 (16.6%) required ICU admission, men were far more prone to need ICU admission, with 74% of those requiring ICU care being male, with an OR = 1.74 and with 95% CI 1.36–2.22 (*p*-value < 0.001) [26]. 

In our analysis, gender was not shown to affect 60-days mortality (*p*-value = 0.491), therefore it was not a variable in the multivariate analysis. What we can assume from our results and from the literature is that males are more prone to severe infection and require ICU admission more often than females, but, when it comes to COVID-19 patients with similar severity in the ICU, gender is not a predictor for worse outcomes.

## 5. Study Limitations

Of the total population included in this retrospective analysis, 96% were unvaccinated. As such, we can assume that the current results concern the unvaccinated population. This is a study limitation because vaccination rates are currently very high in most European countries (>85% of the total population). Another limitation of our study is that this is a single center retrospective study from an ICU in Greece that recruited consecutive patients hospitalized for severe COVID-19 infection. Therefore, these results should be treated with additional consideration in populations that do not resemble the Greek population. Finally, since patients were consecutively included in this retrospective analysis from September 2020 to January 2022, our research lacks the inclusion of patients from the first wave of the pandemic in spring 2020. However, research has shown that the radiologic features of pneumonia attributed to COVID-19 were not found to differ significantly between the first wave of the pandemic and the subsequent ones. Accordingly, our results can be extrapolated to patients currently hospitalized in the ICU due to severe COVID-19, since the major characteristics of severe disease have not drastically altered [27,28].

## 6. Conclusions

The hospitalization of a severely ill patient in the ICU has always been a challenge. When it is combined with a disease such as COVID-19 that is not yet fully understood, every piece of information that can be evaluated should be pursued. The results of our research clearly show that age alone is not a predictor of mortality in severe cases of COVID-19. Instead, it encourages healthcare providers to use more composite scores (such as CCI) in every-day clinical practice in order to evaluate a patient and their chances of survival in the ICU. Will comorbidities and functional status start to change how we think and act in times of crisis? [29]. We hope that our results will provoke skepticism among clinicians and eventually make ageism a less pivotal criterion in decision making.

In times where all effort and focus were targeted on finding new therapeutic choices to combat this new viral threat and towards discovering an effective vaccine, it is crucial not to forget to avoid all other complications in the ICU (with septic complications being the most prevalent). Avoiding bacterial infections and sepsis in the ICU is of utmost importance and can make the difference between life and death in a far more decisive manner than a new antiviral drug could.

## Figures and Tables

**Figure 1 jpm-13-00908-f001:**
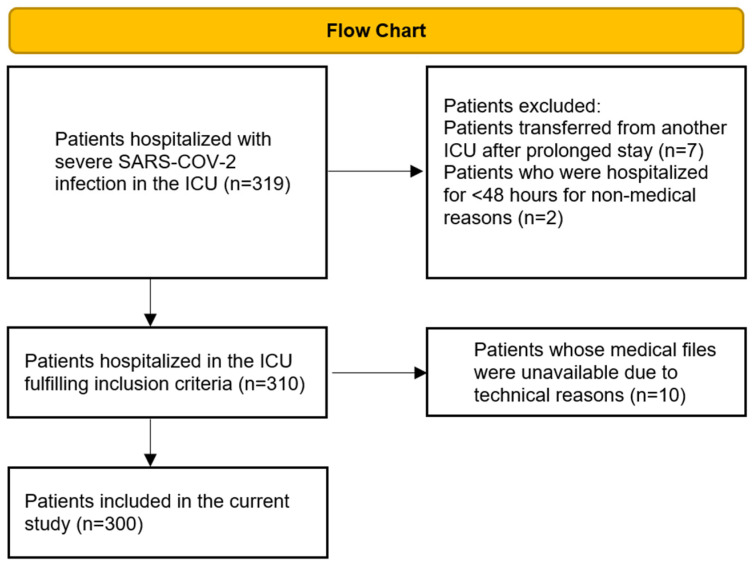
Patient inclusion criteria Flow Chart.

**Figure 2 jpm-13-00908-f002:**
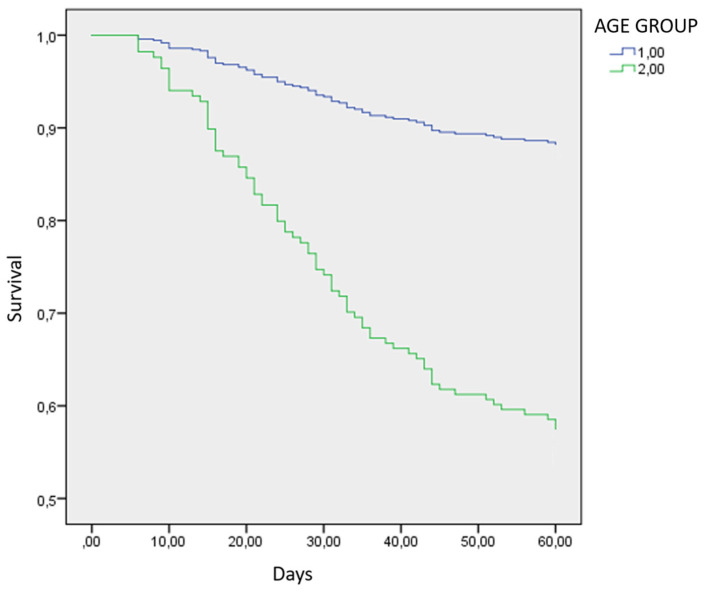
Kaplan–Meier survival curve comparing the two age groups in the univariate analysis. Blue (Group 1): Age group < 65 years old; Green (Group 2): Age group > 65 years old; Age Group (<65 years old) HR: 0.229, 95% CI: 0.140–0.375, *p*-value < 0.001.

**Table 1 jpm-13-00908-t001:** Demographic data of patients.

Characteristics	Total	≥65 Year-Old	<65 Year-Old	*p*-Value
Patient number (n)	300	131	169	
Age (years)	60.4 ± 13.2	72.3 ± 6.1	51.1 ± 9.2	<0.001
Sex Ratio (male-female)	213–87 (71–29%)	88–43 (67–33%)	125–44 (74–26%)	0.199
ΒΜΙ (kg/m^2^)	30.5 ± 6.2	29.8 ± 6.1	31.14 ± 6.1	0.027
Current Smokers (n-%)	42 (14%)	16 (12%)	26 (15%)	0.289
Vaccinated (n-%)	12 (4%)	4 (3%)	8 (5%)	0.439
CCI	2.4 ± 1.9	3.8 ± 1.6	1.3 ± 1.3	<0.001
APACHE II	11.9 ± 4.8	14.3 ± 4.2	10.3 ± 4.4	<0.001
Hypertension (n-%)	136 (45%)	80 (61%)	56 (33%)	<0.001
Diabetes Mellitus (n-%)	67 (22%)	38 (29%)	29 (17%)	0.015
Coronary Artery Disease (n-%)	32 (11%)	21 (16%)	11 (6%)	0.008
Hyperlipidaemia (n-%)	67 (22%)	41 (31%)	26 (15%)	<0.001
COPD (n-%)	45 (15%)	22 (17%)	23 (14%)	0.444
Hypothyroidism (n-%)	46 (15%)	19 (14%)	27 (15%)	0.726

**Table 2 jpm-13-00908-t002:** Univariate-Multivariate Cox Regression—60-days survival.

	Univariate Analysis			Multivariate Analysis		
Variables	HR	95% CI	*p*-Value	HR	95% CI	*p*-Value
60-days Survival						
Age Group (<65 years-old)	0.229	0.140–0.375	<0.001			
Age (constant variant)	1.074	1.053–1.096	<0.001			
Gender (male)	0.840	0.511–1.380	0.491			
CCI	1.674	1.513–1.851	<0.001	1.646	1.247–2.175	<0.001
APACHE II	1.126	1.082–1.172	<0.001			
Not Received Remdesivir	2.500	1.568–3.988	<0.001	2.458	0.980–6.162	0.055
Days of Mechanical Ventilation	1.017	1.011–1.024	<0.001			
Absence of Sepsis	0.067	0.038–0.117	<0.001	0.039	0.013–0.119	<0.001
Days of symptoms	0.925	0.865–0.988	0.021			
d-dimers	1.072	1.032–1.113	<0.001			
CRP	1.026	1.002–1.050	0.031			
NLR	1.028	1.014–1.042	<0.001			
SGOT (AST)	1.000	1.000–1.001	0.010			
SGPT (ALT)	1.001	1.000–1.001	0.019			
Creatinine	2.951	2.288–3.806	<0.001			
Procalcitonin	1.077	1.002–1.157	0.044			
Ferritin	1.000	1.000–1.000	<0.001			

## Data Availability

The data presented in this study are available on request from the corresponding author. The data are not publicly available due to privacy restrictions.

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
