# Peer review of "The Real Impact of Age on Mortality in Critically Ill COVID-19 Patients"

_jpm, 2023, doi:10.3390/jpm13060908_

Round 1
Reviewer 1 Report
Dear Authors,
I would like to thank you for the opportunity of reviewing this interesting paper that is focused on a very remarkable and challenging topic that is a lively argument also in daily clinical practice. The ongoing COVID-19 pandemic has overwhelmed healthcare systems globally, imposing serious health, economic and social effects on the population. It is now established that the likelihood of severe COVID-19 infection increases with age, the presence of comorbidities and the absence of immunization against SARS-COV-2. The aim of the present study was to evaluate the true burden of age on COVID-19 mortality in critically ill patients admitted to ICU.
This paper is pleasurable to read, although it suffers from some limitations that the Authors can easily adjust to slightly improve it, making it more eligible for this important Journal. Furthermore, the Authors can improve some sections of the paper, adding information and including other important references about this topic that, in my opinion, should be cited and discussed.
First of all, although the language used is appropriate, I (I am not a native English speaker) recommend to the Authors obtain a certified native speaker with proficiencies in the scientific-medical field to complete properly this paper (if not yet done). Moreover, I recommend making a further revision of the manuscript to fix some small typing/language errors.
The title is clear and direct. However, personally, I believe it could be improved and more focused on results. For example, “The real impact of age on mortality in critically ill COVID-19 patients” or something like that could probably sound catchier.
Although the introduction fits the context of the study, it is concise. Sometimes, many concepts clearly explicated in an exhaustive introduction could help readers to become passionate about reading the paper and using it as a reference.
Lines 44-45 “It is now established that the likelihood of severe infection increases with age, the presence of comorbidities and of course the absence of immunization against SARS-COV-2”. In my opinion, it is important to underline that despite viral pneumonia has been recognized as the main clinical presentation of this disease and represents the main cause of its severity and mortality, COVID-19 infection can cause several complications due to coagulation disorders and abdominal involvement, especially in severely ill patients and those admitted to ICU [doi:10.3390/diagnostics12040846] [doi: 10.1002/jmv.26294]. Please, cite the aforementioned papers and introduce these important aspects in this section.
Lines 46-56: the same concept is repeated also in the discussion (lines 344-356. Authors should extensively report the results from the COVIP group study either here OR in the discussion. Therefore, one of the two sections should be reduced.
Table 1 is missing some data, for example, the p-values for BMI and APACHE II (or it is wrong formatted). Please check it.
In the Results section, please stick simply to the results. For example, lines 188-190 are not necessary and should be moved to the limitation sections. Similarly, lines 193-194 and lines 263-270 are not necessary. Please check the entire paragraphs and make them sound more scientific, analyzing and commenting on the results only in the following discussion.
Subtitle 3.2.1 “60 days mortality in the ICU predictors” should be removed and the corresponding paragraph merged with the previous one, in order to facilitate Readership comprehension.
The discussion is too long and should be reduced, focusing on the emerging results. Moreover, please limit the use of rhetorical questions. First of all, I suggest analyzing the result regarding the lack of significance of age in predicting survival, which was the primary endpoint of the study. Next, factors that were found to be significant in predicting survival in the study, such as the CCI score and the remdesivir use (which are not currently mentioned in the discussion) and the presence of sepsis, should be addressed. Finally, a brief discussion should be made regarding the other factors, which, unlike other studies, were not statistically significant, such as gender. I also recommend not dividing the discussion into chapters to facilitate Readership comprehension.
Authors should explicit the limitations of their study. For example, they should state that this is a single center study performed exclusively on Greek patients. Moreover, no other concomitant bacterial, fungal or viral infections were investigated as the Authors simply reported the presence of a septic state. This data is essential given that the probability of ICU admission and survival could be influenced by the presence of any other microorganism. Indeed, it is known that ICU patients with COVID-19 often present with a concomitant fungal and bacterial infection [doi: 10.1016/j.chest.2021.04.002] [doi: 10.3390/diagnostics12071617]. If possible, therefore, the Authors should include these data in the analysis. If this data is not available, it is necessary to briefly discuss this topic in the discussion and include this important limitation of the study. Finally, the study period was between September 2020 and January 2022, which does not include the first wave of the pandemic, thus the present results might present an important bias. Despite recent studies have demonstrated that COVID-19 CT pattern prevalence did not statistically differ between the different waves of the pandemic [doi: 10.1007/s10140-021-01937-y.] [doi: 10.7759/cureus.21656], so the results could be deemed enough reliable. Could please the Authors discuss this topic here or in the Discussion section, citing the aforementioned papers?
The conclusion is too long. It should be limited to 4-5 sentences and be more concise. Lines 416-422 and 433-437 can be removed.
Best regards,
Author Response
Thank you for giving us the opportunity to revise our manuscript. We believe that the reviewers’ comments have helped us to improve our work. The point-by-point responses to the reviewers’ comments appear below and have been also added in the marked version of the main manuscript with track changes overview.
Please see the attachment.

Reviewer 2 Report
My specific comments are below-
1. Accordingly, elderly people faced multiple challenges …this is okay. And sometimes were unable to secure a bed in an ICU….I do not understand why this population is different from others. Please revise the statement or make it generalize for the second part of this statement.
2. Detail methodology of sample calculation need to be revealed. How the authors calculated 65 years as upper limit of age for separating two groups. What was the lower limit of age of patients that they included in this study?
3. The first paragraph of introduction is not literally required for this study. It is already reported in many places.
4. It is wise to write sex instead of gender. It is better to present sex ratio in Table 1 rather only male percentage.
5. Conclusion is very large, need to present it in robust and concise way. It can be at best within 150-200 words for this study.
I am not qualified to assess this matter.
Author Response
Thank you for giving us the opportunity to revise our manuscript. We believe that the reviewers’ comments have helped us to improve our work. The point-by-point responses to the reviewers’ comments appear below and have been also added in the marked version of the main manuscript with track changes overview.
Please see attachment.

Round 2
Reviewer 1 Report
The Authors addressed raised points adequately.
Author Response
We would like to thank the reviewer for his insightful comments and his aid in drastically improving the quality of our work.